# Random-effects meta-analysis of effect sizes as a unified framework for gene set analysis

**Mohammad A. Makrooni**[1☯]**, Dónal O'Shea**[1☯]**, Paul Geeleher**[2]**, Cathal Seoighe**[1]*

**1** School of Mathematical and Statistical Sciences, University of Galway, Galway, Ireland, **2** Department of Computational Biology, St. Jude Children's Research Hospital, Memphis, Tennessee, United States of America

☯ These authors contributed equally to this work.
* Cathal.Seoighe@nuigalway.ie

**Data Availability Statement:** All code used in the paper are available at https://github.com/osedo/GSA-MREMA.

**Funding:** CS and MAM are funded by Science Foundation Ireland (https://www.sfi.ie/) under

## Abstract

Gene set analysis (GSA) remains a common step in genome-scale studies because it can reveal insights that are not apparent from results obtained for individual genes. Many different computational tools are applied for GSA, which may be sensitive to different types of signals; however, most methods implicitly test whether there are differences in the distribution of the effect of some experimental condition between genes in gene sets of interest. We have developed a unifying framework for GSA that first fits effect size distributions, and then tests for differences in these distributions between gene sets. These differences can be in the proportions of genes that are perturbed or in the sign or size of the effects. Inspired by statistical meta-analysis, we take into account the uncertainty in effect size estimates by reducing the influence of genes with greater uncertainty on the estimation of distribution parameters. We demonstrate, using simulation and by application to real data, that this approach provides significant gains in performance over existing methods. Furthermore, the statistical tests carried out are defined in terms of effect sizes, rather than the results of prior statistical tests measuring these changes, which leads to improved interpretability and greater robustness to variation in sample sizes.

## Author summary

The role of gene set analysis is to identify groups of genes that are perturbed in a genomics experiment. There are many tools available for this task and they do not all test for the same types of changes. Here we propose a new way to carry out gene set analysis that involves first working out the distribution of the group effect in the gene set and then comparing this distribution to the equivalent distribution in other genes. Tests performed by existing tools for gene set analysis can be related to different comparisons in these distributions of group effects. A unified framework for gene set analysis provides for more explicit null hypotheses against which to test sets of genes for different types of responses to the experimental conditions. These results are more interpretable, because the group effect distributions can be compared visually, providing an indication of how the experimental effect differs between the gene sets.

award 16/IA/4612. DOS is funded by Science Foundation Ireland under award 18/CRT/6214. PG is supported by the NIH, including a K99/R00 award from NHGRI (5R00HG009679-03) and an R35 award from NIGMS (1R35GM138293-01). The funders had no role in study design, data collection and analysis, decision to publish, or preparation of the manuscript.

**Competing interests:** The authors have declared that no competing interests exist.

This is a *PLOS Computational Biology* Methods paper.

## Introduction

Gene set analysis (GSA) methods are used to provide insight into gene expression (or other genomics data types) by testing hypotheses on pre-defined sets of genes. This serves to leverage prior biological knowledge, reduce the number of hypotheses tested and improve the interpretability of the results [1]. Many different GSA methods exist [1], and they can be classified as either competitive or self-contained, based on the null hypothesis being tested [2]. When applied to gene expression data, for example, competitive methods test for enrichment of differentially expressed (DE) genes in gene sets relative to the background. Self-contained methods, on the other hand, assess whether gene sets contain DE genes, without comparing the extent of differential expression to background genes. Methods can be further categorized based on the direction of the expression changes that are the basis of the test. A directional hypothesis involves testing for either up-regulation or down-regulation of genes in a set, while a mixed hypothesis tests for differential expression regardless of direction [3].

The nature of the hypothesized difference between the genes in a set (inset) and the genes outside of a set (outset) provides further method distinctions. For example, in over-representation analysis (ORA) [4] the null hypothesis is that the *proportion* of DE genes in the inset is not greater than the proportion in the outset. Despite being conceptually simple, ORA was found to be the top performing commonly-used method in a review of GSA benchmarking and simulation studies [5]. However, by making a binary classification of genes as either DE or non-DE, based on an arbitrary p-value threshold, ORA discards information about the extent of the difference in expression and is tied to the power of the specific experiment performed. Instead of making a binary classification, gene set enrichment analysis (GSEA) [6] ranks genes by a ranking metric and then tests against a null hypothesis under which the rank distribution of a gene set is independent of the sample group. Various ranking metrics have been used in many similar methods [1]. These methods have the advantage over ORA of taking the full gene rank into account, which reflects the size of the DE effect. Although metrics can be used that are influenced by both the effect size estimate and its uncertainty (such as the signal-to-noise ratio option in GSEA [7]), these must be collapsed into a single value for ranking purposes, resulting in a loss of information. The use of the signal-to-noise ratio in GSEA can also create difficulties for the inclusion of covariates and interaction terms in the experimental design [8]. Failure to account for these covariates could lead to spurious results in some cases and reduced power to detect true enrichment in others. Ranking-based methods typically evaluate the significance of the differences between gene sets through sample permutation. This has the benefit of making the methods robust to failures in the assumptions (such as gene-gene independence) that are made by other methods, but at the expense of being computationally expensive and not suited to experiments with low sample numbers [9].

Here, we propose a novel approach to GSA that both provides a unifying framework for the different approaches outlined above and also takes into account the uncertainty in the estimate of the effect size from the differential expression analysis. In our approach, the log fold change (LFC) for genes in a given set is modeled as a mixture of Gaussian distributions, with distinct components corresponding to up-regulated, down-regulated and non-DE genes. We use the Expectation Maximization (EM) algorithm to estimate the parameters of this mixture distribution. Using a methodology inspired by statistical meta-analysis [10], the standard error of the DE effect size estimate is incorporated into the estimation procedure, with genes with large

standard errors having less influence on the parameter estimates than genes for which the DE effect is estimated with greater precision. A wide range of tests that are relevant for gene set analysis can be performed by applying model comparison techniques to estimated effect size distributions in different gene sets. We evaluated the performance of our approach relative to existing GSA implementations using both simulated data as well as real data. Our method showed substantially increased power compared to existing methods in the simulations. When applied to real data from The Cancer Genome Atlas (TCGA), our approach showed more power than established methods in calling cancer-associated gene sets as enriched for DE genes across different cancer types. In a separate collection of gene expression datasets associated with human diseases, which has been used previously to evaluate the performance of GSA methods, our approach tended to give a higher rank than other methods to the disease-associated gene sets within the corresponding experiment.

## Results

### Model and implementation

We model the LFC of genes as a mixture of three Gaussian components, corresponding to non-DE, up-regulated and down-regulated genes and make use of random effects meta-analysis to incorporate the variance in the gene-level LFC estimates in parameter estimation (see Methods for details). The statistical tests at the gene set level consist of comparison of the fit of models in which the LFC distribution is the same (null hypothesis) or different (alternative hypothesis) for genes in the inset (the gene set of interest) and outset (the remaining genes in the background set). We evaluated three forms of model comparison, corresponding to different kinds of GSA. The first consists of comparing the fits of a model in which all parameters of the LFC mixture distribution are shared between genes in the inset and outset to a model that allows the proportion of DE genes to be larger in the inset. We refer to this as the one degree of freedom (1DF) test. It evaluates whether there is a greater proportion of DE genes in the inset than in the outset. The second kind of test evaluated allows both the proportion of DE genes and the fraction of the DE genes that are upregulated/downregulated to differ between the inset and the outset, making it a two degrees of freedom (2DF) test. In the third kind of test, we compared the fit of a model in which all parameters of the mixture distribution are allowed to differ between genes in the inset and outset to a model in which all parameters are shared. This test has six degrees of freedom (referred to as the 6DF test) and is sensitive to any differences in the mixture distribution between the inset and outset (e.g. a difference in the size of the DE effect even if the proportion of DE genes is the same). In all cases we used the Likelihood Ratio test (LRT) for the comparison of model fit (details of all of the tests are provided in Methods).

### Performance on simulated data

We first used simulations to assess method performance relative to existing methods. In the simulations the group effect was assumed to be normally distributed in the inset and outset, but with a larger variance in the inset, resulting in a larger number of DE genes with LFC above the threshold shown (Fig 1A). This is deliberately far-removed from our model, which fits a mixture of three normal distributions. The simulations illustrate a key feature that distinguishes our approach to GSA from many existing methods. GSA methods designed to identify gene sets with a higher proportion of DE genes typically perform statistical tests on the proportions of genes falling below an arbitrary p-value threshold. This proportion, however, will depend on the number of samples in an experiment. Indeed, it is likely to be the case that all genes are DE to some extent between any two biologically distinct groups of samples, with this

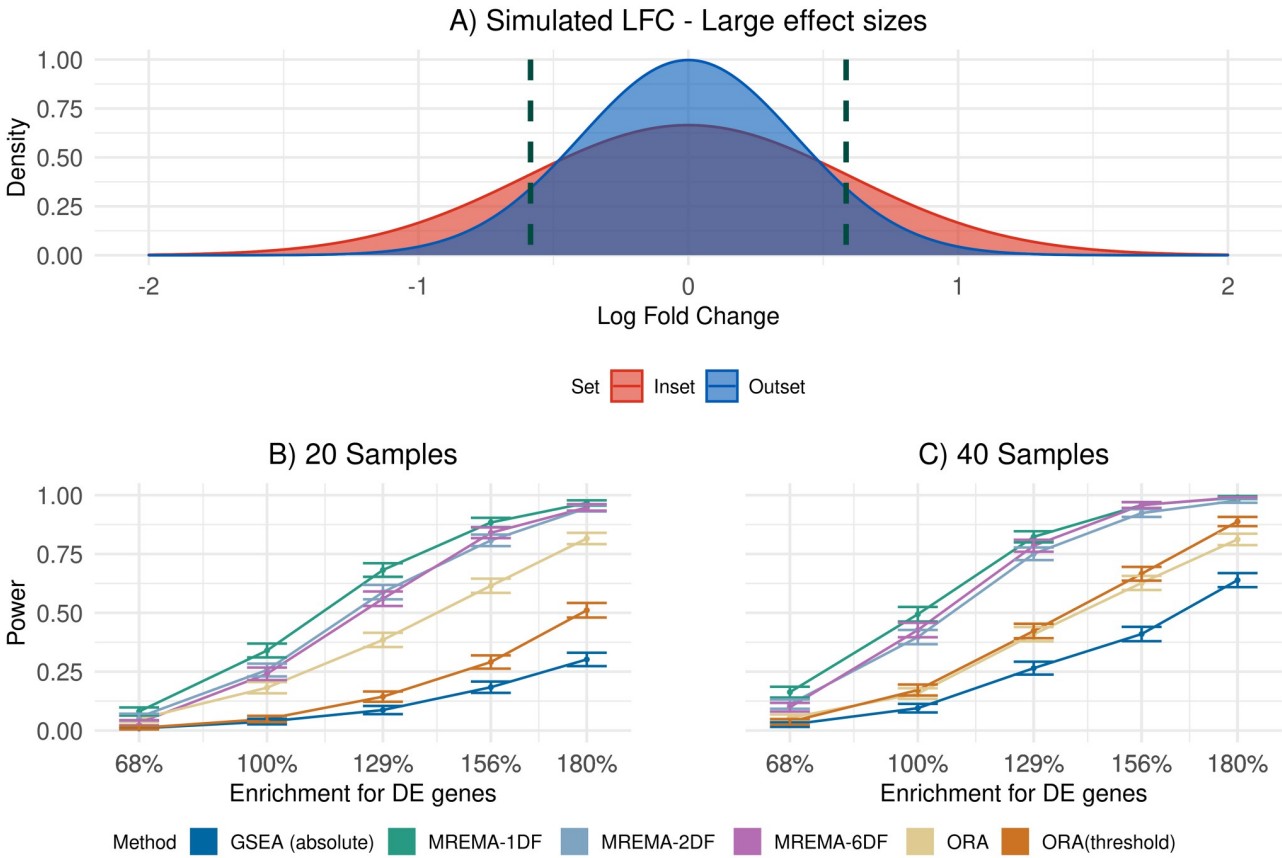

**Fig 1. Power on simulated data with large effect sizes.** A) LFC distributions in the inset (red) and outset (blue) for an example simulation. The variance of the inset distribution was allowed to change, leading to simulations with different levels of enrichment for DE genes in the inset. The dashed lines show the location of the threshold (fold-change of 1.5) used to define differential expression for methods that apply such a threshold. B, C) Comparison of power of different methods as a function of the level of enrichment in the inset for simulations consisting of 20 (B) or 40 (C) samples. The x-axis indicates the relative enrichment in the inset of the proportion of the LFC distribution beyond the upper or lower thresholds.

difference becoming statistically significant, given sufficient sample numbers. Instead, we define DE by setting a threshold on the absolute value of the LFC and our method compares proportions of genes for which the true (but unknown) absolute value of the LFC is above this threshold.

Our method provided a significant improvement in power, defined as the proportion of enriched gene sets detected, compared to ORA, and GSEA on the simulated data (Figs 1B, 1C, 2B and 2C). As expected, the improvement in power was sensitive to the threshold used to define differential expression (S1, S2 and S3 Figs). The undirectional version of GSEA, using the absolute signal to noise ratio (SNR) as the ranking metric, had low power with low sample numbers, but its power increased with increasing sample numbers, due to the increased number of sample permutations available to derive the p-value. Accounting for the uncertainty in the gene-level effect size estimates in our method resulted in a significant improvement in power, relative to a version of our method that did not take the standard error of the effect-size estimates into account (Fig 3). As expected, this improvement was greater for lower sample numbers, for which there is greater uncertainty in the effect size estimates. The false positive rate of our method was similar to that of existing methods (S4 Fig) and it remained below the significance threshold whether or not the standard error of the effect size estimate was included in the model (S5 Fig).

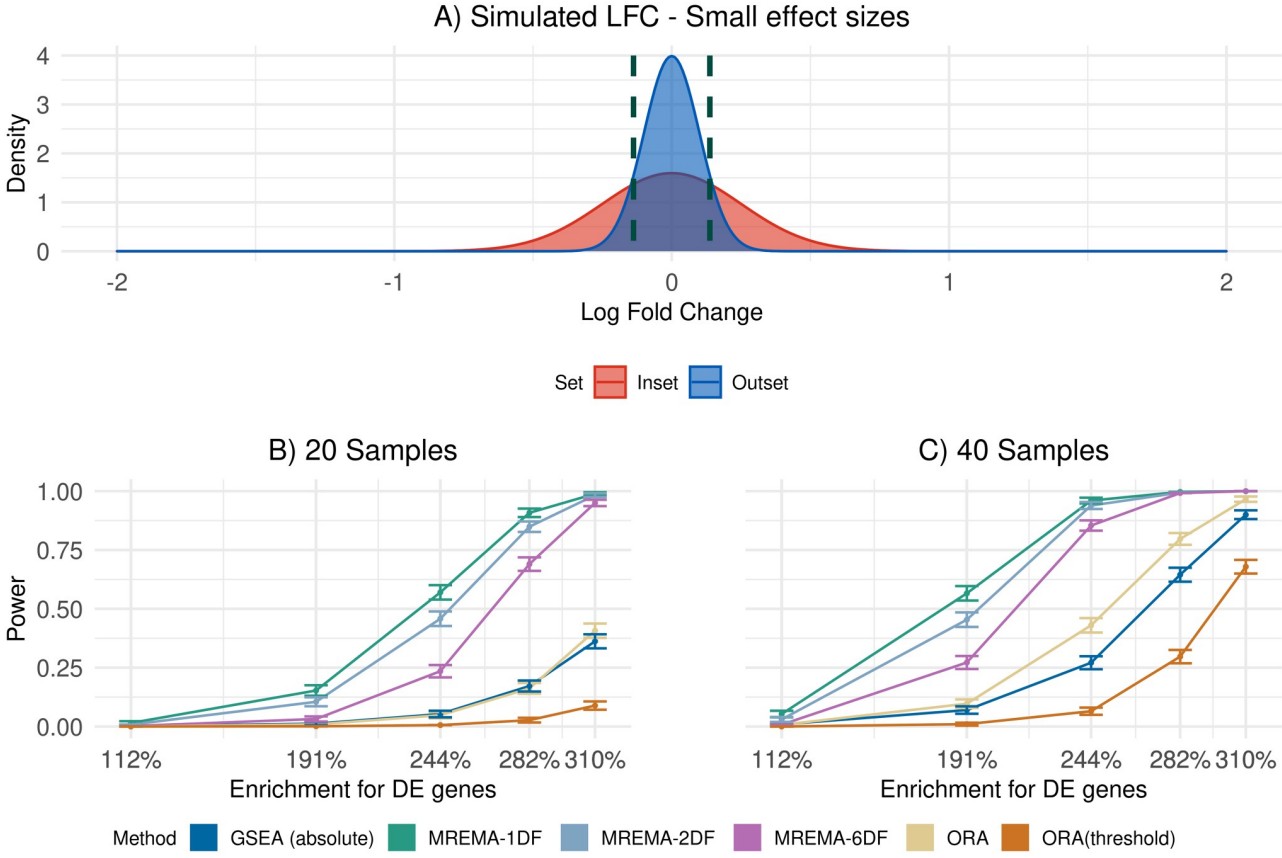

**Fig 2. Power on simulated data with small effect sizes.** A) LFC distributions in the inset (red) and outset (blue) for an example simulation. As in ([Fig 1]) the variance of the inset distribution was allowed to change, leading to simulations with different levels of enrichment for DE genes in the inset. The dashed lines show the location of the threshold (in this case a fold-change of 1.1) used to define differential expression. B, C) Sample sizes of 20 (B) or 40 (C) were simulated. The x-axis indicates the relative enrichment in the inset of the proportion of the LFC distribution beyond the upper or lower thresholds.

Our method requires researchers to specify upfront the magnitude of the DE effects of interest. Unsurprisingly, the power of our method to detect the difference in the proportion of genes above the threshold in the simulated data depended on the relationship between the DE threshold and the LFC distributions in the inset and outset (Figs 1 and 2 and S1–S3 Figs). The highest power is achieved when the threshold defines a region in the effect size distribution in which there is a substantial difference between the inset and outset. No such fold-change threshold is specified for existing enrichment-based methods and, instead, such methods often rely on p-value thresholds from statistical tests at the individual gene level. The lack of an explicit definition of differential expression can lead to statistical inconsistencies. For example, the power of ORA decreased with increasing sample numbers in some simulation settings (Fig 1 and S6 Fig). This resulted from an increase in the power to detect the smaller expression changes in the outset as the sample number increases, causing a reduction in the difference in proportions of significantly differentially expressed genes. A threshold can also be applied in the original differential expression analysis when using ORA, with the test for enrichment applied to the proportion of genes that are statistically significant and have absolute LFC above the threshold. This resulted in a consistent increase in power in ORA with increasing sample numbers (Figs 1 and 2 and S1–S3 Figs), highlighting the importance of explicitly defining the effect sizes that are of interest.

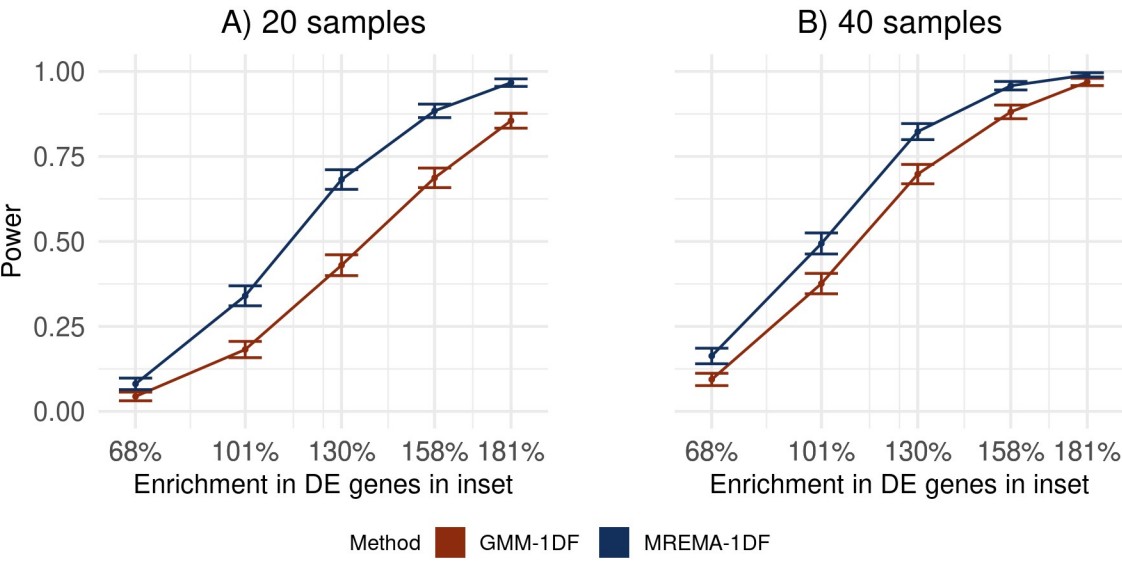

**Fig 3. Incorporating uncertainty boosts power.** Power obtained with and without taking account of the standard error of the gene-level effect size estimates is shown in blue and red, respectively for simulations based on 20 (A) or 40 (B) samples. The parameter space of the simulations was the same as in Fig 1.

The improvement in power over established methods was maintained in simulations that included unequal numbers of upregulated and downregulated genes (S7 Fig). The power of the more constrained undirectional 1DF test and ORA was unchanged as the proportion of upregulation among the DE genes varied. As expected, the directional GSEA achieved high power once a majority of DE genes in a set were upregulated. Our 2DF test and 6DF test also increased in power in these scenarios, as they had greater flexibility to fit the differences in the effect size distributions between the inset and outset. These simulations consisted of 10,000 genes, 100 of which were in the enriched gene set. We also assessed the effect of varying the gene set size on power. Unsurprisingly, power decreased for all methods as the gene set size was reduced (S8 Fig). When the size of the enriched set was increased to 200 genes, power improved for all methods. In all scenarios investigated our approaches maintained a power advantage over established methods. Unlike other methods, the 6DF test is specifically designed to identify differences in DE effect size distributions between genes in the inset and outset, regardless of whether the proportion of DE genes is different. In simulations consisting of a difference in effect sizes, but not in the proportion of DE genes between the inset and outset, the 6DF test was sensitive to this difference, whereas the other methods were not (S6 Fig).

## Real data analysis

We tested 347 KEGG gene sets for enrichment of DE between tumour and normal samples using data from 15 cancer types obtained from TCGA [11]. The differential expression analysis and benchmarking of our results was carried out using GSEABenchmarkeR [3]. Our approaches had a far lower runtime than the permutation-based GSEA, while the runtime of ORA was very low in comparison to our approach (S9 Fig). All of our approaches found a higher proportion of significant gene sets than either GSEA, directional GSEA or ORA, before and after correcting for multiple testing (S10 Fig). The low proportion of significant gene sets, after correcting for multiple testing is consistent with results reported in previous benchmarking studies [3]. In order to evaluate the power of our approach we compared the distribution

of the nominal p-values of cancer associated gene sets across all cancer types, with the p-values found by established methods (Fig 4). Our approach compared favorably (i.e. tended to have lower p-values for cancer-associated gene sets) with established methods across the four cancer-associated gene sets. We also obtained the rank for each gene set (among the 347 gene sets tested) in each cancer type. All of the gene sets evaluated tended to be ranked higher using our approach than by existing methods (Fig 4A, 4B and 4C), with the exception of the miRNAs in cancer gene set (Fig 4D).

We examined the ranking of specific disease-associated gene sets in 38 microarray datasets corresponding to these diseases. These datasets have previously been compiled and used to evaluate gene set analysis tools [12] [13] [3]. Our methods performed well on these data, particularly for very highly ranked gene sets (Fig 5). For example 21% of these gene sets were ranked in the top ten using our 1DF and 2DF tests and 16% for the 6DF test; however, only 5% and 8% were in the top 10 using GSEA and and ORA, respectively. The 1DF maintained an advantage over other methods for a wider range of values of the number of top-ranked gene sets considered, but the performance of all methods became closer as the number of top-ranked genes considered became very large (S11 Fig). This was expected, as increasing the number of top-ranked genes to consider allowed more methods to recover the relevant disease-associated gene set, even when it was not highly ranked.

## Discussion

Many existing GSA methods test hypotheses that are defined in terms of the results of previous statistical tests, carried out on individual genes, rather than on the actual effect of interest. For example, the null hypothesis of enrichment-based methods (such as ORA) is that the proportion of genes passing the p-value significance threshold is the same in the inset and outset. This proportion depends on the power of the experiment and is liable to change in both sets if the experiment is repeated with a different number of samples, sometimes in unpredictable ways (Fig 1). Fundamentally, therefore, existing GSA methods provide results that relate to the specific experiment performed, rather than to the biological effects of interest. We propose an alternative approach to GSA in which the null hypotheses evaluated are expressed in terms of the underlying biology. Other approaches test whether the proportion or distribution of statistically significant genes differs between the inset and outset. We on the other hand assess whether the proportion of genes, for which the true (but unknown) DE effect exceeds some specified threshold, differs between the gene sets. This enables the researcher to consider the minimal effect size that may be of biological significance, rather than focusing on whether individual genes are statistically significant in a specific experiment.

The three tests we evaluated here consider distinct null hypotheses. The 1DF test evaluates whether the proportion of DE genes differs between the inset and outset. The 2DF test evaluates whether the proportion of upregulated or downregulated genes differs between the inset and outset, while the 6DF test is sensitive to differences in the magnitude of the effect sizes as well as the variation in the proportions of DE genes. While the goal of the 1DF and 2DF tests is similar to that of existing enrichment-based GSA methods, the fold-change value at which a gene is considered to be DE is not explicitly defined in ORA or in many other established GSA methods (although tools such as DESeq2 allow for LFC thresholds to be set when calling a gene as DE [14]). This is an important parameter that should be considered in the context of established methods as well, in order to reduce the influence of DE effects which, though statistically significant, are too small to be biologically meaningful. For example, in the extreme case of an infinite number of samples, there will be no error in the LFC estimates and all genes with non-zero fold-change will be statistically significant. Our approach involves setting a DE

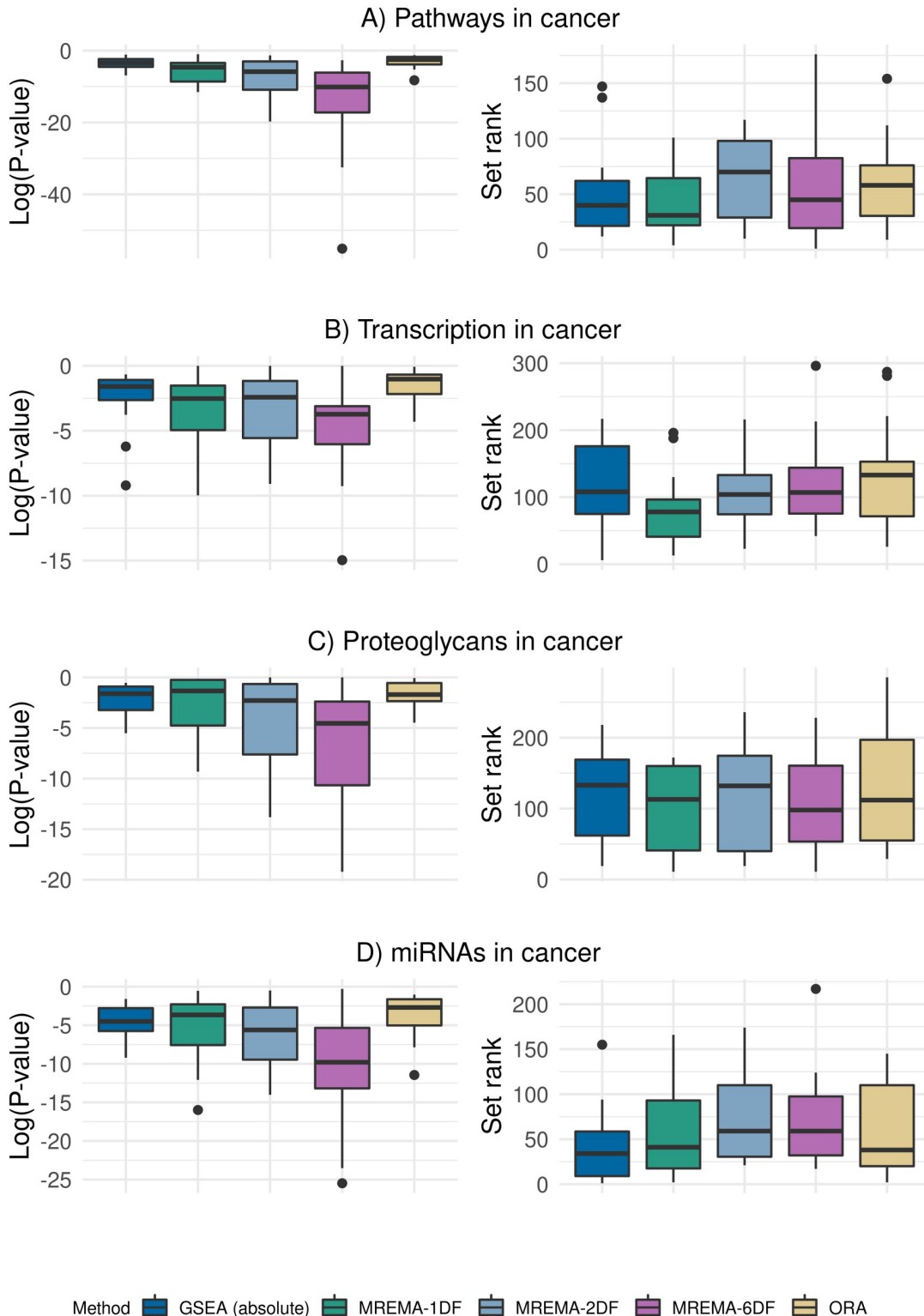

**Fig 4. Enrichment for cancer-associated gene sets in TCGA data.** Boxplots of log-transformed nominal p-values across 15 cancer types are shown on the left column (lower values indicate more significant results). The rank of the same cancer-associated gene sets is shown in the right-hand column (lower values indicate higher rank). Each row shows the results obtained for the gene set named above the row. A fold-change threshold of 1.5 was used for the MREMA tests.

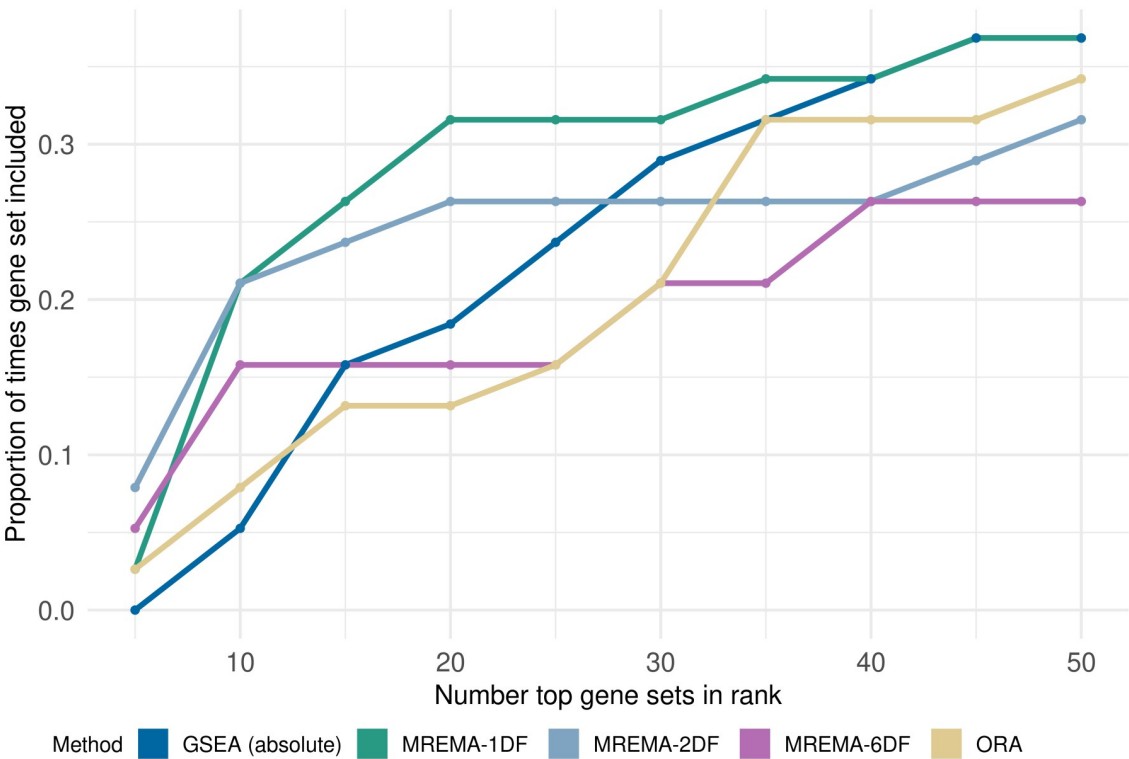

**Fig 5. Ranking of disease-associated gene sets in corresponding gene expression experiments.** The proportion of times the disease-associated gene set was included in the *x* top-ranked gene sets as a function of *x*. The MREMA tests had higher values of this proportion than established methods when only the most highly ranked gene sets were considered (i.e. low values of *x*, corresponding to the left-hand side of the plot). The 1DF test maintained this advantage for longer than the other two tests, with GSEA matching the performance of the 1DF test by the right-hand side of the plot. A fold-change threshold of 1.5 was used for the MREMA tests. The position of the disease-associated sets in the full ranking is available in supplementary data (S1 Table).

threshold in advance, enabling the researcher to define the effect sizes of interest and leading to more interpretable results. The 1DF and 2DF tests are similar in that they both test for enrichment in the proportion of DE genes; however, the 2DF test is sensitive to the direction of the effect and can have more power when the proportion of up-regulated genes among the DE genes differs between the inset and outset (S7 Fig). The null hypothesis for the 6DF test is that the LFC distributions for genes in the inset and outset are the same. This is a more general test that is sensitive to differences in both the proportion of affected genes and the magnitude of the effect. The results can be made more interpretable by visualizing the inferred effect size distribution, which can reveal why the null hypothesis is rejected for a specific gene set. Given that GSEA considers the full distribution of gene ranks, rejection of its null hypothesis can result from differences in proportions of DE genes or differences in the effect sizes of the DE genes or both. This is comparable to the 6DF test, though the reason for the rejection of the null hypothesis may be much less clear.

A further distinction between our approach to GSA and existing methods is that we take account of the uncertainty in the estimated group effect for each gene. Existing methods, by contrast, typically obtain a p-value or test statistic from an initial gene expression analysis and either count the number of genes with p-values below some threshold or combine the uncertainty with the effect size in a ranking metric. By considering both the estimated effect and its standard error our method retains information about the size of the effect and the precision with which it has been estimated. This information is then used to estimate the parameters of a

distribution describing the effect size in a set of genes, with the standard error of the estimate being used to reduce the influence on the parameters of this distribution of effects that are poorly estimated. Taking account of this uncertainty resulted in an increase in power (Fig 3). As expected, this improvement is greater for lower sample numbers where there is greater uncertainty in the effect size estimates. In this study we used a Gaussian mixture model (GMM) for the distribution of the logarithm of gene expression fold-change (LFC) between groups, because this allows the uncertainty in the effect size estimate to be incorporated easily into the estimation of the component variances within the M step of the EM algorithm. In principal, the GMM could be replaced with a more general model for the distribution of effects across genes, but this would require the development of efficient methods to incorporate the standard error of the effect sizes into the parameter estimation.

Our method can have a significant advantage over methods that are based on detecting differences in proportions of genes passing a p-value threshold using the hypergeometric distribution, particularly in the presence of relatively modest sample sizes or modest group effects that are shared across many genes in a gene set. This is because our method can accumulate evidence of a difference in the group effect distribution between gene sets, even when there is large uncertainty in the group effect for individual genes. In such a case, the proportion of DE genes passing the p-value threshold may be small, corresponding to low power to detect enrichment of moderately DE genes. By contrast, where the DE status of a gene is clear, the advantage to avoiding hard assignment of genes as either DE or not DE is smaller. Consistent with this, our method retained power well with smaller sample numbers (Figs 1 and 2).

Methods, such as GSEA, that are based on sample permutation have the advantage of robustness, particularly to independence assumptions [7]. For example, ORA treats genes as independent observations, even though genes can share regulatory mechanisms, resulting in correlated expression across samples. This robustness comes at a cost, however, as sample permutation methods are computationally intensive and not practical for small sample numbers (S9 Fig). The interpretation of results can also be challenging. The null hypothesis for GSEA is that the observed enrichment score for a gene set is not a result of differences between cases and controls. Rejection of this null hypothesis does not necessarily imply that genes within the gene set are affected disproportionately by the sample group, as the group effect is absent altogether in the permuted samples. Our approach can also be adapted to use sample permutation, leading to greater robustness, but maintaining an advantage in terms of interpretability. For example, a probabilistic estimate of the proportion of genes in a gene set with group effect above a specified threshold can be obtained by summing over the gene-specific effect-size estimates (treating each effect size estimate as a random variable). A permutation-based test for a difference in this proportion between genes in the inset and outset can then be obtained by shuffling the sample labels. However, rerunning the full DE analysis for each permutation would be very computationally expensive using existing DE analysis tools. Another direction of future development could be to downweight genes that contribute to multiple gene sets as is done in PADOG to take account of overlapping membership between gene sets [12]. Our approach already weights genes inversely by the uncertainty in the group effect size estimate. It should be possible to modify this weight to take account also of the number of gene sets to which a gene contributes, but this will require further work to ensure validity and to test the accuracy of the results obtained.

When applied to real data our approach showed significant improvements in runtime compared to permutation-based approaches (S9 Fig). After correcting for multiple testing our 1DF, 2DF and 6DF approaches found approximately 12%, 25% and 37%, respectively, of the gene sets to be significant while established methods are left with very few significant gene sets (S10 Fig). For the 6DF test this may seem like a high proportion of significant sets, but this

approach tests for any differences in the LFC distribution, not just differences in weight assigned to the DE components. It is plausible that many gene sets will differ from the background in some respect, especially when comparing tumour and normal samples. Our three approaches returned more significant nominal p-values than established methods for cancer-associated gene sets, across the different cancer types (Fig 4). The 6DF test consistently returned more significant p-values, as allowing the magnitude of LFC to differ between the inset and outset, increases the range of signals to which the method is sensitive. Our ranking of these cancer-associated sets also compared well with established methods (Fig 4A, 4B and 4C), with the 1DF test tending to show an advantage over the 2DF test and the 6DF test. Our approach also appeared to show improved results when compared to established methods for ranking specific disease associated gene sets in the relevant disease datasets (Fig 5). The improvement over established methods was most noticeable in the higher ranks of the results (i.e. towards the left hand side of Fig 5), where our 1DF and 2DF tests identified the disease-associated gene sets more frequently than other methods. This is significant as the highest-ranked genes tend to receive the most attention in gene set analysis.

The choice of test depends on the interests of the researchers and should be decided a priori. If the objective is to identify gene sets enriched for DE genes without regard to the magnitude of the effect, then the 1DF test or 2DF test is most appropriate. The 2DF test will tend to have more power than the 1DF if the gene set is primarily upregulated or primarily downregulated, whereas the 1DF test will have more power if the proportions of upregulated and downregulated genes in the gene set are similar in the inset and outset. If there is an interest in whether the gene set may be enriched for particularly large effects, in addition to differences in the proportion of affected genes then the 6DF test is appropriate.

## Conclusion

We describe a unified framework for GSA that supports the formulation and testing of different types of hypotheses relating to how genes with a shared annotation respond to an experimental condition of interest. Our approach can also be used to carry out a post-hoc analysis of enriched gene sets, providing information on *how* the gene set compares to other genes and not just on *whether* the gene set is enriched for genes that are perturbed under the experimental condition. We propose an approach to GSA based on this unified framework that can evaluate the range of hypothesis tests that are implicit in established GSA methods. Simulations suggest that this approach can provide increased power relative to established methods. When applied to real data our approaches showed promising results, ranking putatively relevant gene sets more highly than established methods.

## Materials and methods

### Gaussian mixture model (GMM)

We model the LFC using a mixture of Gaussian distributions, one with a mean of zero, for non-DE genes, and two with positive and negative means to capture positive and negative DE genes respectively. The LFC distribution is then given by:

$$\sum_{l=1}^{3} \pi_l N(\mu_l, \sigma_l^2) \quad (H0) \tag{1}$$

where $\mu_l$, $\sigma_l^2$ and $\pi_l$ are the mean, variance and weight, respectively, of component $l$. The weights of all components sum to one. As a null hypothesis, we use a shared mixture distribution for all genes, regardless of gene set membership. In the alternative models gene set

membership is included in the LFC distribution as follows:

$$\sum_{l=1}^{3} z_i \pi_{1l} N(\mu_{1l}, \sigma_{1l}^2) + \sum_{l=1}^{3} (1 - z_i) \pi_{2l} N(\mu_{2l}, \sigma_{2l}^2) \quad (H1) \tag{2}$$

where $z_i$ is a binary variable indicating the gene set membership of gene $i$. This allows the LFC distribution to differ between the inset and the outset. In the alternative model for the 1DF test (S1 Appendix Eq. 29), all genes contribute to the estimate of the means, variance and a parameter, $c$, which determines the proportion of DE genes that are upregulated in both the inset and outset (S1 Appendix Eq. 30). The weight assigned to the non-DE genes in the inset is estimated using only genes in the set and the weight assigned to the non-DE genes in the outset is estimated using only genes outside the set. This means that while the proportions of DE genes can differ between the inset and the outset the fraction of those DE genes that are upregulated/downregulated is the same. In the alternative model for the 2DF test both the proportion of DE genes and the fraction of those genes that are upregulated/downregulated are estimated separately for the inset and outset (S1 Appendix Eq. 21). In the alternative model for the 6DF test all parameters for the inset distribution are estimated using the genes in the set while all parameters in the outset distribution are estimated using genes outside the set (S1 Appendix Eq. 3).

Our method uses a soft threshold, $\tau$, to define biologically relevant DE. We place the constraint on the parameters of the mixture components corresponding to upregulated and downregulated genes such that at most 0.25 of the area of either component falls in the interval $[-\tau, +\tau]$. In each M-step of the EM algorithm a minimum value is set for the mean of the upregulated component to satisfy this condition. If the mean of the upregulated component is estimated to be below the minimum value, the minimum value replaces the estimate. This minimum value is then updated in the next iteration of the EM algorithm. A dynamic maximum value for the mean of the downregulated component is likewise used to replace any estimated mean that leads to 0.25 of the area of that component above $\tau$. The variance of the non-DE component is fixed such that 95% of the area of the component lies within the interval $[-\tau, +\tau]$.

## Mixture random-effects meta-analysis

The GMM described above does not consider the standard error associated with the LFC estimate for each gene. Each estimated LFC comes with uncertainty, in other words each estimated LFC is the sum of the true, but unknown, LFC, and an error term. We assume normally distributed error with mean zero and standard deviation given by the standard error of the estimated LFC ($\sigma_i$, for gene, $i$). Compounding the GMM and the error we obtain the following LFC distribution:

$$\sum_{l=1}^{3} \pi_l N(\mu_l, \sigma_l^2 + \sigma_i^2) \quad (H0) \tag{3}$$

Similar to the previous section, we can build the alternative model by adding the binary indicator for the set membership:

$$\sum_{l=1}^{3} z_i \pi_{1l} N(\mu_{1l}, \sigma_{1l}^2 + \sigma_i^2) + \sum_{l=1}^{3} (1 - z_i) \pi_{2l} N(\mu_{2l}, \sigma_{2l}^2 + \sigma_i^2) \quad (H1) \tag{4}$$

Considering each gene as a single study and interpreting $\sigma_i^2$ as the within study variance and $\sigma_l^2$ as between study variance, we see that our model is similar to standard random effect

meta analysis except that the underlying distribution of LFC is a GMM instead of a single distribution. To estimate the parameters of the model we use the Expectation Maximization algorithm. In the M-step we use an iterative approach, optimizing successively over the mean and variance parameters of each Gaussian distribution in the mixture model. This is because unlike a simple GMM, the contribution of each gene to the parameters is weighted not only by the posterior probability (calculated in the E-step) but also by a weight whose formula is given by [10]. Additional model and implementation details are provided in S1 Appendix.

## Simulations

We simulated RNA-Seq gene expression data for *N* individuals in two equally sized groups. The genes were divided into S non-overlapping gene sets of equal size. We sampled gene expression counts for each gene, *i*, from the negative binomial distribution $NB(\mu_i, \varphi_i)$. In order to choose realistic parameters for this distribution, we used a real data set. The mean and the dispersion parameters for all genes in this data set were estimated using DESeq2 and the BRCA-TCGA data. For each gene in the count data matrix, we randomly chose a mean and a dispersion parameter from these values. The expression values for non-DE genes were again sampled from $NB(\mu_i, \varphi_i)$ and expression values for DE genes were sampled from $NB(\mu_i, \varphi_i)$ for the control group. The expression values for the case group was sampled from $NB(FC\mu_i, \varphi_i)$ distribution, with *FC* the fold change value.

For the main power simulations (Figs 1 and 2), LFC values for the inset and the outset were drawn from a normal distribution with a mean of zero and the standard deviation differing between the inset and the outset. Equal numbers of DE genes were up- and down-regulated in the inset and outset. In each power simulation there was 10,000 genes split into 100 gene sets of 100 genes each. All gene sets were tested for enrichment and p-values were corrected for multiple testing using the Benjamini-Hochberg procedure. We applied a significance threshold of 0.05 on the adjusted p-values. In order to highlight significantly enriched gene sets we imposed a second condition, as well as statistical significance, the estimated proportion of DE genes had to be higher in the gene set than in the background. This ensured that we avoided identifying gene sets that were significantly depleted for DE genes. For the null simulations (S4 Fig) the LFC values of all sets were drawn from the same normal distribution. Each of the 100 gene sets were tested for enrichment. This was repeated 10 times to give 1,000 tests of enrichment where none was present. We also performed simulations with unequal fractions of upregulated/downregulated genes in the inset and outset (for the results shown in S6, S7 and S8 Figs). The LFC values for these simulations were drawn from a mixture of three Gaussian distributions (corresponding to upregulated and downregulated genes and a component centred on zero for non-DE genes). Using DESeq2 [14] we obtained an estimate of the LFC and the standard error of this estimate. Established methods were implemented using the *GSEABenchmarkeR* package in R [3], with the exception of GSEA(absolute) which was run using the absolute signal to noise ratio as the ranking metric [15]. Both GSEA and GSEA(absolute) were run with 1,000 permutations.

## Real data

Gene count data for 15 TCGA paired tumour and normal datsets, and KEGG gene sets were retrieved using the *GSEABenchmarkeR* package [3]. Differential expression analysis was again carried out using DESeq2. The established methods were implemented as in the simulations. A fold-change threshold of 1.5 was used for the 1DF, 2DF and 6DF tests. The same approach was used for the microarray data, with the exception that the differential expression analysis was carried out using *limma* [16]. The results of the MREMA tests were ranked as follows for

both the TCGA and microarray data. For all gene sets where the estimated proportion of DE genes was higher in the set than the background the ranking was according to p-value (in ascending order). This placed sets with the highest statistical significance for enrichment at the top. Below the enriched sets, gene sets that were assigned a lower proportion of DE genes than the background were ranked by p-value in ascending order. This meant that gene sets that were significantly depleted were ranked towards the bottom.

## Supporting information

**S1 Fig. Power simulations—Using a fold-change threshold of 1.1.** The power for the different tests is shown using a FC threshold of 1.1. The power for high LFC values illustrated in A) is shown in B) for 20 samples and C) for 40 samples. The power for low LFC values, illustrated in D) is shown in E) for 20 samples and F) for 40 samples.
(TIF)

**S2 Fig. Power simulations—Using a fold-change threshold of 1.3.** The power for the different tests is shown using a FC threshold of 1.3. The power for high LFC values illustrated in A) is shown in B) for 20 samples and C) for 40 samples. The power for low LFC values, illustrated in D) is shown in E) for 20 samples and F) for 40 samples.
(TIF)

**S3 Fig. Power simulations—Using a fold-change threshold of 1.5.** The power for the different tests is shown using a FC threshold of 1.5. The power for high LFC values illustrated in A) is shown in B) for 20 samples and C) for 40 samples. The power for low LFC values, illustrated in D) is shown in E) for 20 samples and F) for 40 samples.
(TIF)

**S4 Fig. False-positive rate in simulated data.** The false positive rate in null simulations with large effect sizes for A) 20 samples and B) 40 samples. The false positive rate in null simulations with small effect sizes for C) 20 samples and D) 40 samples. For our tests a gene set was deemed a false positive if the nominal p-value was less than 0.05 and the proportion of DE genes was estimated to be higher in the gene set than in the background.
(TIF)

**S5 Fig. False-positive rate in simulated data, with and without accounting for uncertainty.** The false positive rate of the 1DF test with (blue) and without (red) accounting for the uncertainty in the LFC estimates.
(TIF)

**S6 Fig. Power simulations—Difference between inset and outset distributions with no change in proportion of DE genes.** The power for all approaches as the LFC distribution changes without increasing the proportion of genes above a threshold of 1.5, for A) 20 samples and B) 40 samples.
(TIF)

**S7 Fig. Power simulations—Different proportions of upregulated/downregulated genes in inset and outset.** The power for all approaches when the DE genes are A & D) equally split between upregulated and downregulated, B & E) 75% upregulated and C & F) 100% upregulated.
(TIF)

**S8 Fig. Power simulations—Different gene set sizes.** The power for all approaches to identify enrichment illustrated in red in A) as the size of the enriched gene increases from B) 25 genes

to C) 50 genes to D) 100 genes to E) 200 genes.
(TIF)

**S9 Fig. Runtime across fifteen cancer datasets.**
(TIF)

**S10 Fig. Proportion of significant gene sets across fifteen different cancer types.** The proportion of A) nominally significant gene sets across fifteen cancer datasets and B) significant gene sets after correcting for multiple testing using the BH procedure. For our tests a gene set was deemed significant if the p-value was less than 0.05 and the proportion of DE genes was estimated to be higher in the gene set than in the background.
(TIF)

**S11 Fig. Full ranking of disease-associated sets.** The proportion of times the disease-associated gene set was included in the $x$ top-ranked gene sets as a function of $x$. This is equivalent to Fig 5, but showing the entire range of values of $x$.
(TIF)

**S1 Appendix. Methods—Mixture of Random Effect Meta-Analysis (MREMA).**
(PDF)

**S1 Table. Position of disease associated gene sets in ranking.**
(XLSX)

## Acknowledgments

The results published or shown here are in whole or part based upon data generated by the TCGA Research Network: https://www.cancer.gov/tcga.

## Author Contributions

**Conceptualization:** Cathal Seoighe.

**Data curation:** Dónal O'Shea.

**Formal analysis:** Mohammad A. Makrooni, Dónal O'Shea.

**Funding acquisition:** Cathal Seoighe.

**Investigation:** Dónal O'Shea.

**Methodology:** Mohammad A. Makrooni, Cathal Seoighe.

**Software:** Dónal O'Shea.

**Supervision:** Paul Geeleher, Cathal Seoighe.

**Visualization:** Dónal O'Shea.

**Writing – original draft:** Dónal O'Shea.

**Writing – review & editing:** Mohammad A. Makrooni, Paul Geeleher, Cathal Seoighe.

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
