## [Decision Letter · Decision Letter 0]

30 Jun 2022

Dear Prof. Seoighe,

Thank you very much for submitting your manuscript "Random-effects meta-analysis of effect sizes as a unified framework for gene set analysis" for consideration at PLOS Computational Biology.

As with all papers reviewed by the journal, your manuscript was reviewed by members of the editorial board and by several independent reviewers. In light of the reviews (below this email), we would like to invite the resubmission of a significantly-revised version that takes into account the reviewers' comments.

The three reviewers have provided many questions regarding algorithmic and implementation details, simulation details and real application details. Addressing these questions will be essential to improve the manuscript.

We cannot make any decision about publication until we have seen the revised manuscript and your response to the reviewers' comments. Your revised manuscript is also likely to be sent to reviewers for further evaluation.

Sincerely,

Jie Liu

Associate Editor

PLOS Computational Biology

Jian Ma

Deputy Editor

PLOS Computational Biology

The three reviewers have provided many questions regarding algorithmic and implementation details, simulation details and real application details. Addressing these questions will be essential to improve the manuscript.

Reviewer's Responses to Questions

**Comments to the Authors:**

Reviewer #1: Review is uploaded as an attachment.

Reviewer #2: Makrooni et al. proposed to build a gaussian mixture model (GMM) to better quantify the effect size of differentially expressed (DE) genes and the proportion of DE genes in the Gene Set Analysis (GSA). Such a GMM strategy includes an error term in estimating the Log-Fold-Change (LFC) variance for each gene to extract the LFC estimation uncertainty, boosting the gene set detection power. The authors showed the advantages of their MREMA model over the standard GSA methods through simulation and real data analysis from the TCGA datasets. Competing with the widely used GSEA method is challenging. Despite lacking a few mathematical and implementation details of the model, this paper makes me feel concerned about the persuasion to the potential users with one set of simulations (Figure1C&D) and one specific pathway detection in the TCGA-BLCA study (Figure3).

Please find the attachment for more detailed questions.

Reviewer #3: Please see my comments attached

**Have the authors made all data and (if applicable) computational code underlying the findings in their manuscript fully available?**

Reviewer #1: Yes

Reviewer #2: Yes

Reviewer #3: **No: **The Github page did not provide the example data to run the code

PLOS authors have the option to publish the peer review history of their article (what does this mean?). If published, this will include your full peer review and any attached files.

Reviewer #1: No

Reviewer #2: No

Reviewer #3: No
---

## [Decision Letter · Decision Letter 1]

12 Sep 2022

Dear Prof. Seoighe,

Thank you very much for submitting your manuscript "Random-effects meta-analysis of effect sizes as a unified framework for gene set analysis" for consideration at PLOS Computational Biology. As with all papers reviewed by the journal, your manuscript was reviewed by members of the editorial board and by several independent reviewers. The reviewers appreciated the attention to an important topic. Based on the reviews, we are likely to accept this manuscript for publication, providing that you modify the manuscript according to the review recommendations.

Sincerely,

Jie Liu

Academic Editor

PLOS Computational Biology

Jian Ma

Section Editor

PLOS Computational Biology

[LINK]

Reviewer's Responses to Questions

**Comments to the Authors:**

Reviewer #1: The authors addressed my comments and suggestions. I only have a few more minor comments as follows.

1. Line 17 the literature number for ORA is missing.

2. In Fig.1B and Fig.1C, it's hard to distinguish ORA (threshold) and MERMA-6DF and the colors are too similar.

3. Fig.3 is mentioned before Fig.2 (line 112)

4. Fig.1 and Fig.2 large/small efect sizes, does it mean large/small log fold change values?

5. Fig.S1 shows that if we use a small threshold for LFC when the LFC are large, ORA actually has the best power. The authors should provide some explanation about this. Is it because ORA has higher FPR (Fig. S4 cd) when the null simulations contains high LFC? Also, I would suggest the authors to switch the order of Fig.S4 (ab) with Fig.S4(cd) to be consistent with the order of Fig.S1/Fig.S2 and Fig.1/Fig.2, where the results show in order of high/low LFC respectively.

6. I would suggest using the name in Fig.S5 as MERMA-1DF instead of GMM to be consistent with other plots.

Reviewer #2: This revised version gives me much more confidence. I only have a few very minor suggestions:

1. Authors need to put more effort into polishing manuscript writing. For example, Fig. 2 is cited after Fig. 3 . A LOT of figure citations do not appear within the parenthesis (e.g., lines 105, 116, 154 etc.). There are still a few long, complicated and fragmental sentences. Those are very hard to follow (e.g., lines 196-199).

2. Methods comparison figures are not consistent. Some figures contain more or fewer methods (e.g., Supplementary Figures 1-3 do not have GSEA. Supplementary Figure 7 adds GSEA and GSEA (absolute) ). Authors should make the methods to be compared and the color settings consistent for easy comparison.

3. GitHub software also needs further polishing to include the parameter explanation, input data format introduction, and output data interpretation/explanation.

Reviewer #3: The review is uploaded as an attachment

**Have the authors made all data and (if applicable) computational code underlying the findings in their manuscript fully available?**

Reviewer #1: Yes

Reviewer #2: Yes

Reviewer #3: Yes

PLOS authors have the option to publish the peer review history of their article (what does this mean?). If published, this will include your full peer review and any attached files.

Reviewer #1: No

Reviewer #2: No

Reviewer #3: No

Figure Files:

Data Requirements:

Reproducibility:

References:

---

## [Decision Letter · Decision Letter 2]

18 Sep 2022

Dear Prof. Seoighe,

We are pleased to inform you that your manuscript 'Random-effects meta-analysis of effect sizes as a unified framework for gene set analysis' has been provisionally accepted for publication in PLOS Computational Biology.

Best regards,

Jie Liu

Academic Editor

PLOS Computational Biology

Jian Ma

Section Editor

PLOS Computational Biology

Reviewer's Responses to Questions

**Comments to the Authors:**

Reviewer #1: The authors have addressed all my comments. I don't have any further comments.

Reviewer #2: Authors have addressed my questions

Reviewer #3: The authors have addressed all the comments.

**Have the authors made all data and (if applicable) computational code underlying the findings in their manuscript fully available?**

Reviewer #1: Yes

Reviewer #2: None

Reviewer #3: Yes

PLOS authors have the option to publish the peer review history of their article (what does this mean?). If published, this will include your full peer review and any attached files.

Reviewer #1: No

Reviewer #2: No

Reviewer #3: No

---

## [Editor Report · Acceptance letter]

29 Sep 2022

PCOMPBIOL-D-22-00849R2 

Random-effects meta-analysis of effect sizes as a unified framework for gene set analysis

Dear Dr Seoighe,

I am pleased to inform you that your manuscript has been formally accepted for publication in PLOS Computational Biology. Your manuscript is now with our production department and you will be notified of the publication date in due course.

With kind regards,

Zsofi Zombor
